# Resistance exercise protects mice from protein-induced fat accretion

**Michaela E Trautman**[1,2,3], **Leah N Braucher**[1,2], **Christian Elliehausen**[1,2,4], **Wenyuan G Zhu**[1,2,5], **Esther Zelenovskiy**[1,2], **Madelyn Green**[1,2], **Michelle M Sonsalla**[1,2,5], **Chung-Yang Yeh**[1,2], **Troy A Hornberger**[5,6], **Adam R Konopka**[1,2,4], **Dudley W Lamming**[1,2,3,4,5,7]\*

[1]Department of Medicine, University of Wisconsin-Madison, Madison, United States; [2]William S. Middleton Memorial Veterans Hospital, Madison, United States; [3]Nutrition and Metabolism Graduate Program, University of Wisconsin- Madison, Madison, United States; [4]Cellular and Molecular Biology Graduate Program, University of Wisconsin-Madison, Madison, United States; [5]Comparative Biomedical Sciences Graduate Program, University of Wisconsin-Madison, Madison, United States; [6]School of Veterinary Medicine, University of Wisconsin-Madison, Madison, United States; [7]University of Wisconsin Carbone Cancer Center, Madison, United States

*For correspondence:
dlamming@medicine.wisc.edu

**Abstract** Low-protein (LP) diets extend the lifespan of diverse species and are associated with improved metabolic health in both rodents and humans. Paradoxically, many athletes and body-builders consume high-protein (HP) diets and protein supplements, yet are both fit and metabolically healthy. Here, we examine this paradox using weight pulling, a validated progressive resistance exercise training regimen, in mice fed either an LP diet or an isocaloric HP diet. We find that despite having lower food consumption than the LP group, HP-fed mice gain significantly more fat mass than LP-fed mice when not exercising, while weight pulling protected HP-fed mice from this excess fat accretion. The HP diet augmented exercise-induced hypertrophy of the forearm flexor complex, and weight pulling ability increased more rapidly in the exercised HP-fed mice. Surprisingly, exercise did not protect from HP-induced changes in glycemic control. Our results confirm that HP diets can augment muscle hypertrophy and accelerate strength gain induced by resistance exercise without negative effects on fat mass, and also demonstrate that LP diets may be advantageous in the sedentary. Our results highlight the need to consider both dietary composition and activity, not simply calories, when taking a precision nutrition approach to health.

## eLife assessment

This study presents a **valuable** finding on the relationship between high-protein diet and resistance exercise on fat accumulation and glucose homeostasis. The evidence supporting the claims of the authors is **solid**, although the inclusion of mechanistic insight would have strengthened the study. The work will be of interest to dietician and exercise biologists working to understand the synergy between diet and physical activity.

## Introduction

In contrast to the long-standing idea that 'a calorie is a calorie,' research over the last two decades has shown that calories from different macronutrient sources have distinct impacts on health (*Hall et al., 2015*). Dietary protein in particular has been shown to have a critical role in the regulation of metabolic health and longevity; low-protein (LP) diets are associated with reduced rates of diabetes and

other age-related diseases in human longitudinal studies, and promote leanness and insulin sensitivity in human randomized clinical trials of metabolically unhealthy adults (*Ferraz-Bannitz et al., 2022*; *Fontana et al., 2016*; *Levine et al., 2014*; *Sluijs et al., 2010*). In mice, LP, high carbohydrate diets reduce adiposity, improve glucose tolerance, and extend lifespan (*Hill et al., 2022*; *Richardson et al., 2021*; *Solon-Biet et al., 2014*; *Solon-Biet et al., 2015*).

The idea that LP diets are beneficial – and high-protein (HP) diets are deleterious – challenges conventional wisdom. Several HP 'fad' weight loss diets have been popularized over the last two decades, and many researchers and medical professionals support consuming more protein to support healthy aging (*Rodriguez, 2015*). In addition to promoting satiety, some short-term studies have found that HP intake improves glucose control in adults with type 2 diabetes (*Dong et al., 2013*; *Gannon et al., 2003*; *Seino et al., 1983*), and HP diets are recommended for physically active individuals to support muscle growth and strength improvements (*Andersen et al., 2005*; *Willoughby et al., 2007*). Both exercise and amino acids activate the mechanistic target of TOR (mTOR) protein kinase, which stimulates the protein synthesis machinery needed to stimulate skeletal muscle hypertrophy (*Schiaffino et al., 2021*; *Simcox and Lamming, 2022*). To support this process, the Academy of Nutrition and Dietetics recommends consuming 1.2–2.0 g of protein per kg of body weight (BW) per day in physically active individuals (*Thomas et al., 2016*). These protein estimations are substantially higher than the Recommended Daily Allowance for sedentary people of 0.8 g/kg BW, an amount intended to provide sufficient nutrients in 97.5% of people, but it remains controversial if protein intake above this level is detrimental in this population.

If high dietary protein intake was always deleterious, the many individuals consuming HP diets or taking widely sold protein or BCAA supplements to maximize muscle growth following resistance training would have higher adiposity and an increased risk of diabetes. However, there is no data available to support such a surprising conclusion, and indeed the opposite is true: exercise reduces the risk of developing type 2 diabetes (*Grøntved et al., 2012*). We are therefore left with a paradox: while the human and animal data suggesting that increased dietary protein intake is detrimental for metabolic health and increases the risk for numerous age-related diseases is robust, some of the people consuming the highest levels of protein – athletes – are metabolically very healthy.

Here, we examine the possibility that exercise can protect mice from the deleterious metabolic effects of an HP diet normally observed in sedentary mice. In order to evaluate the metabolic effects of exercise and dietary protein intake independent of weight loss, we utilized a recently validated method of progressive resistance exercise for mice that does not cause weight loss (*Zhu et al., 2021*) to assess the interaction of dietary protein content and exercise in male C57BL/6J mice. We fed mice either an LP (7% of calories from protein) or an HP (36% of calories from protein) diet, and either pulled an increasing load of weight down a track 3× per week for 3 mo, or pulled a sham unloaded cart. During the course of the experiment, we comprehensively assessed weight and body composition, metabolic parameters, and the fitness of the mice on these regimens. Finally, we euthanized the mice to determine the effect of both diet and exercise on tissue weight and muscle mass, muscle fiber-type cross-sectional area and distribution, and muscle mitochondrial respiration.

In agreement with our hypothesis that HP diets impair metabolic health in sedentary mice, we found that HP-fed sham-exercised mice gained excess fat mass and increased in adiposity relative to LP-fed mice. In sharp contrast, HP-fed mice that engaged in resistance exercise showed hypertrophy of specific muscles and were protected from fat accretion. However, exercise did not protect HP-fed mice from the effects of protein on blood sugar control. Interestingly, while exercising HP-fed mice gained strength more rapidly than exercising LP-fed mice, the difference in maximum weight that could be pulled by each mouse was not significant, but both the HP diet and weight training regimen increased muscle diameter and in general, size. Interestingly, neither diet nor exercise altered mitochondrial respiratory capacity. Our research shows that resistance exercise protects from HP-induced increases in adiposity in mice and suggests that metabolically unhealthy sedentary individuals consuming an HP diet or protein supplements might benefit from either reducing their protein intake or beginning a resistance exercise program.

# Results

## Resistance exercise protects against HP-induced fat and body weight gain

Seven-week-old C57BL/6J mice were randomized to groups of equal weight and body composition, and placed on either an LP (7% of calories from protein) or HP (36% of calories from protein) diet. Of note, while the protein content of rodent chow varies, typically 17–24% of calories are derived from protein (*Tuck et al., 2020*). The two diets were isocaloric as calories from fat were kept at 19%, and carbohydrates were reduced in the HP diet to compensate for the increased calories from amino acids. We have previously utilized this same LP diet, and the full composition of both diets can be found in *Table 1*. Throughout the course of the study, weight was collected weekly and body composition was analyzed every 3 wk via EchoMRI (*Figure 1A*). Starting after 5 wk on the diets, we exercised half of the mice on each diet using a recently validated mouse model of resistance exercise training, weight pulling (WP), in which mice pull progressively heavier weights down a track; the other half of the mice were sham-exercised by pulling an unloaded cart.

On both a per-animal and a weight-normalized basis, LP diet-fed mice ate more than HP-fed mice, consistent with the satiating effect of dietary protein (*Figure 1B and C*). There was no effect of WP on food consumption (*Figure 1B and C*). Despite consuming fewer calories than LP-fed mice – the two diets have the same caloric density – HP-fed mice gained more weight over the course of 18 wk. By the end of the experiment, HP-fed WP mice had gained significantly less weight than HP-fed sham-exercised controls (*Figure 1D*). As we expected based on previous studies investigating the effects of an LP diet as well as data demonstrating the beneficial effects of protein on muscle growth (*Andersen et al., 2005*; *Willoughby et al., 2007*), HP-fed mice gained significantly more lean mass than LP-fed mice, although surprisingly exercise did not affect this gain (*Figure 1E*). In accordance with our previous results, we found that sham-exercised HP-fed mice gained substantially more body fat than their LP-fed counterparts; however, we found that WP was protective against these effects (*Figure 1F*).

At the conclusion of the experiment, we euthanized the animals and measured the weight of several fat depots and the liver. We found that the inguinal white adipose tissue (iWAT) and epididymal white adipose tissue (eWAT) were significantly heavier in sham-exercised HP-fed mice than in WP HP-fed mice (*Figure 1G and H*). While there was no statistically significant effect on the mass of brown adipose tissue (BAT), the weight of this depot also trended higher in sham-exercised HP-fed mice than in WP HP-fed mice (*Figure 1I*). Finally, there was a statistically significant effect of diet on liver weight, with HP-fed mice having heavier livers, an effect that was larger in sham-exercised mice than in WP mice (*Figure 1J*).

We collected portions of the liver and iWAT for Oil-Red-O and H&E staining, respectively, and quantified the size and number of hepatic lipid droplets and adipocytes. Despite the increased liver weight of the HP groups, they did not have increased lipid accumulation; instead, there was a nonsignificant trend toward both larger and a greater number of liver lipid droplets in the LP-fed groups, with no effect of WP (*Figure 2A–C*). There was no difference in average iWAT adipocyte size or number (*Figure 2D–F*).

## Mice fed an LP diet have better glycemic control and higher energy expenditure than HP-fed mice

We and others have previously shown that consumption of an LP diet improves glucose tolerance and insulin sensitivity, while the literature suggests that resistance exercise improves insulin sensitivity in mice and humans (*Cui et al., 2020*; *Fontana et al., 2016*; *Green et al., 2022*; *Kullmann et al., 2022*; *Laeger et al., 2014*; *Mcleod et al., 2019*; *Westcott, 2012*; *Yu et al., 2021*). We performed glucose and insulin tolerance tests; in agreement with our previous findings, we found that LP-fed mice had improved glucose tolerance relative to HP-fed mice, and there was a trend (p=0.0788) toward a positive effect of WP on glucose tolerance (*Figure 3A*). Similarly, while LP-fed mice were more insulin-sensitive than HP-fed mice under sham conditions, we did not observe a difference between LP and HP-fed mice that performed WP (*Figure 3B*).

In agreement with a model in which high dietary protein is deleterious to sedentary animals, we observed that HP-fed mice had higher fasting blood glucose than LP-fed mice under sham exercise

**Table 1.** Diet composition.

| Amino acid-defined diets | Low protein | High protein |
|---|---|---|
| Teklad diet name | 7% protein calories | 36% protein calories |
| Teklad diet number | TD.140712 | TD.220097 |
| Color | Blue | Green |
| Formula | g/kg | g/kg |
| Sucrose | 291.248 | 214.867 |
| Corn starch | 232.4 | 110.7 |
| Maltodextrin | 232.4 | 110.7 |
| Corn oil | 52.0 | 52.0 |
| Olive oil | 29.0 | 29.0 |
| Cellulose | 30.0 | 30.0 |
| Mineral mix, AIN-93G-MX (94046) | 35.0 | 35.0 |
| Calcium phosphate, dibasic | 8.2 | 8.2 |
| Vitamin mix, Teklad (40060) | 10.0 | 10.0 |
| % kcal from | | |
| Protein | 7.1 | 36.4 |
| Carbohydrate | 74.4 | 44.7 |
| Fat | 18.5 | 18.9 |
| kcal/g | 3.9 | 3.9 |
| Amino acid profile | g/kg | g/kg |
| L-Lysine HCl | 6.64 | 33.308 |
| L-Methionine | 2.18 | 10.95 |
| L-Cystine | 2.34 | 11.767 |
| L-Arginine | 2.05 | 10.296 |
| L-Phenylalanine | 2.15 | 10.787 |
| L-Tyrosine | 2.25 | 11.277 |
| L-Histidine HCl, monohydrate | 1.15 | 7.518 |
| L-Isoleucine | 2.54 | 12.748 |
| L-Leucine | 8.27 | 41.512 |
| L-Threonine | 3.16 | 15.853 |
| L-Tryptophan | 1.1 | 5.557 |
| L-Valine | 2.735 | 13.729 |
| L-Aspartic acid | 6.7 | 33.634 |
| L-Glutamic acid | 9.43 | 47.347 |
| L-Alanine | 3.05 | 15.33 |
| Glycine | 0.96 | 4.838 |
| L-Proline | 2.41 | 12.111 |
| L-Serine | 2.41 | 12.111 |

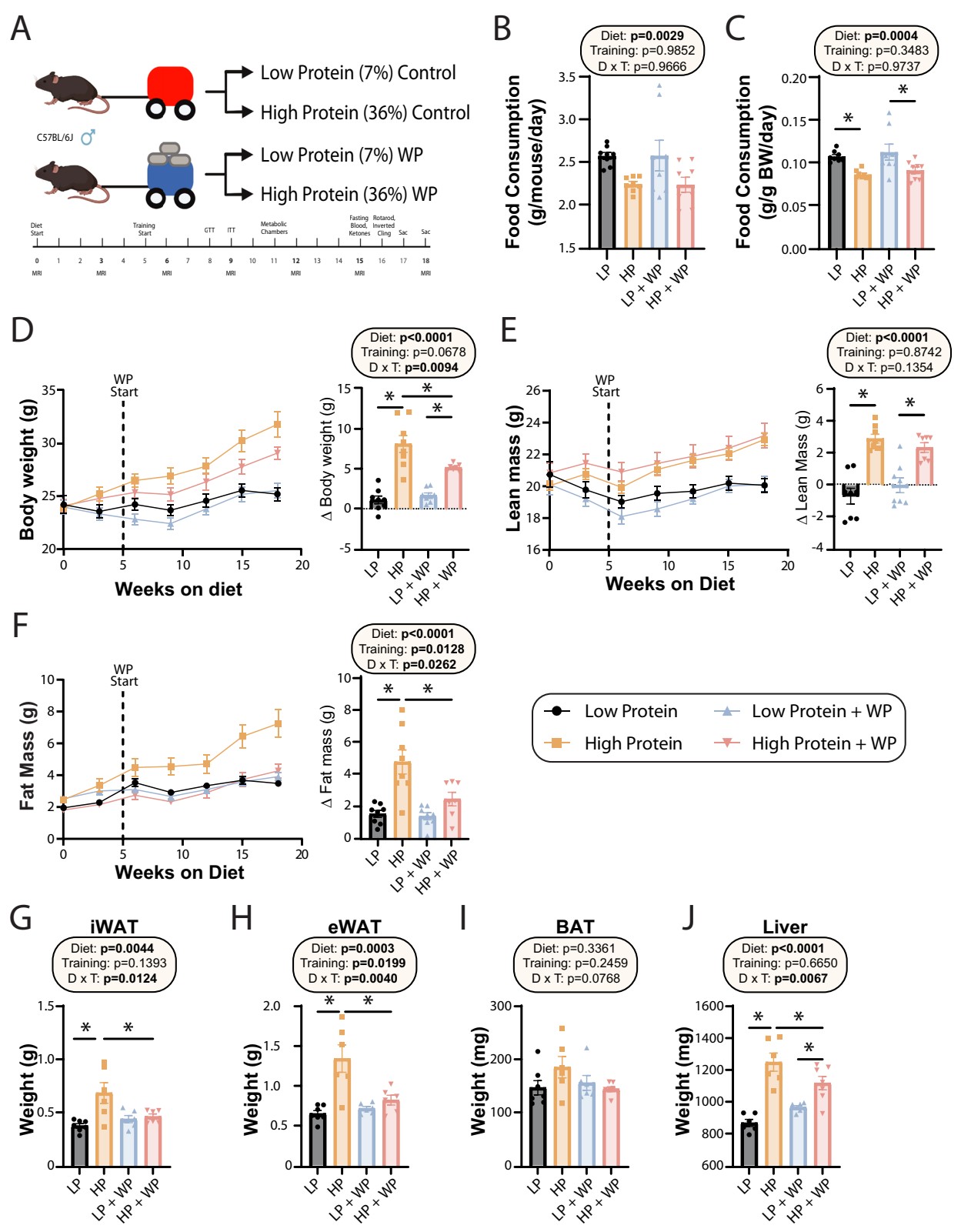

**Figure 1.** Weight pulling protects from high-protein diet-induced weight and fat gain. (**A**) Experimental design. (**B, C**) Food consumption per mouse (**B**) or normalized to body weight (**C**) after ~6 wk on the indicated diets. n = 8/group. (**D–F**) Body weight (**D**), lean mass (**E**), and fat mass (**F**) over time, and change (Δ) from the beginning to end of study. n = 7–8 mice/group. (**G–J**) Weight of the inguinal white adipose tissue (iWAT) (**G**), epididymal white adipose tissue (eWAT) (**H**), brown adipose tissue (BAT) (**I**), and liver (**J**) at the conclusion of the study. n = 6–7 mice per group. (**B–J**) Statistics for the

*Figure 1 continued on next page*

*Figure 1 continued*

overall effects of diet, training, and the interaction represent the p-value from a two-way ANOVA; *p<0.05, Sidak's post-test examining the effect of parameters identified as significant in the two-way ANOVA. Data represented as mean ± SEM.

The online version of this article includes the following source data for figure 1:

**Source data 1.** Weight pulling protects from high-protein diet-induced weight and fat gain.

conditions (*Figure 3C*). Interestingly, WP did not correct this difference – instead, we saw that there was an overall effect of HP-feeding toward increased fasting blood glucose level in both sham-exercised and WP mice (*Figure 3C*). A similar effect of diet was noted on fasting insulin levels, with HP-fed WP mice having significantly higher insulin levels than LP-fed WP mice (*Figure 3D*). In agreement with the overall effect of diet on fasting glucose and insulin levels, we also observed an overall effect of diet on insulin sensitivity calculated via the HOMA2-IR method, with LP WP mice having significantly better insulin sensitivity than HP-fed WP mice (*Figure 3E*).

We and others have previously shown that an LP diet increases fasting FGF21 levels in C57BL/6J male mice (*Green et al., 2022*; *Hill et al., 2022*), and we determined that LP-fed sham mice had significantly higher levels of FGF21 in their blood compared to their HP-fed counterparts (*Figure 3F*). Interestingly, WP mice had a strong overall trend (p=0.0751) toward lower FGF21 levels, and LP-fed WP mice did not have significantly higher FGF21 levels than HP-fed WP mice (*Figure 3F*). FGF21 promotes ketogenesis, and in agreement with our FGF21 data we observed an overall effect of diet on fasting blood ketone levels, with LP-fed mice having higher ketone levels than HP-fed mice (*Figure 3G*). Intriguingly, there was a significant interaction of diet and exercise, with WP lowering ketone levels in HP-fed mice.

Finally, we examined the effect of diet and exercise on blood lipids. Some studies have shown altered lipid levels in mice and humans fed an LP diet (*Treviño-Villarreal et al., 2018*), and we considered it likely that WP would promote a healthy lipid profile. Surprisingly, while diet had no effect on triglyceride or lipid levels, there was an overall increase of triglycerides in WP-fed mice, which was statistically significant in HP-fed mice (*Figure 3H*). There was no effect of diet or exercise on blood levels of cholesterol (*Figure 3I*).

In order to learn more about the effects of the HP diet on weight and adipose gain, we used metabolic chambers to evaluate multiple components of energy balance, including food consumption, spontaneous activity, and energy expenditure. In agreement with our prior home-cage observations and the well-known satiating effect of dietary protein, we observed that HP-fed mice tended to consume less food than LP-fed mice, although this did not reach statistical significance (*Figure 3—figure supplement 1A*). There was no overall effect of exercise on spontaneous activity, but in the light phase there was an overall effect of diet, with LP-fed mice moving more (*Figure 3—figure supplement 1B*).

We examined respiratory exchange ratio (RER) by calculating the ratio of $O_2$ consumed and $CO_2$ produced; an RER of close to 1.0 indicates that carbohydrates are being preferentially utilized for energy production, while a value near 0.7 indicates that lipids are the predominant energy source (*Bruss et al., 2010*; *Yu et al., 2019*). As we anticipated, there was an overall effect of diet on RER, with mice consuming the LP diet having a higher RER during the dark cycle, reaching statistical significance in the WP groups (*Figure 3—figure supplement 1C and D*). In agreement with previous studies, we observed a significant effect of the LP diet on energy expenditure, both on a per-animal basis and when weight was considered as a covariate (*Figure 3—figure supplement 1E and F*). Resistance exercise did not have an overall effect on either RER or energy expenditure.

## Training while consuming an LP diet does not compromise maximum strength or coordination

Prior dogma suggests that in order to maximize strength and muscle hypertrophy from resistance exercise, consumption of a large quantity of high-quality protein is necessary (*Phillips et al., 2005*; *Thomas et al., 2016*). In agreement with this, we observed an significant overall effect of diet on the maximum load mice were able to pull during their training sessions as well as the corresponding area under the curve (AUC) (*Figure 4A*), consistent with HP-fed mice being able to pull more weight. However, the difference was smaller than we had anticipated and was statistically significant only

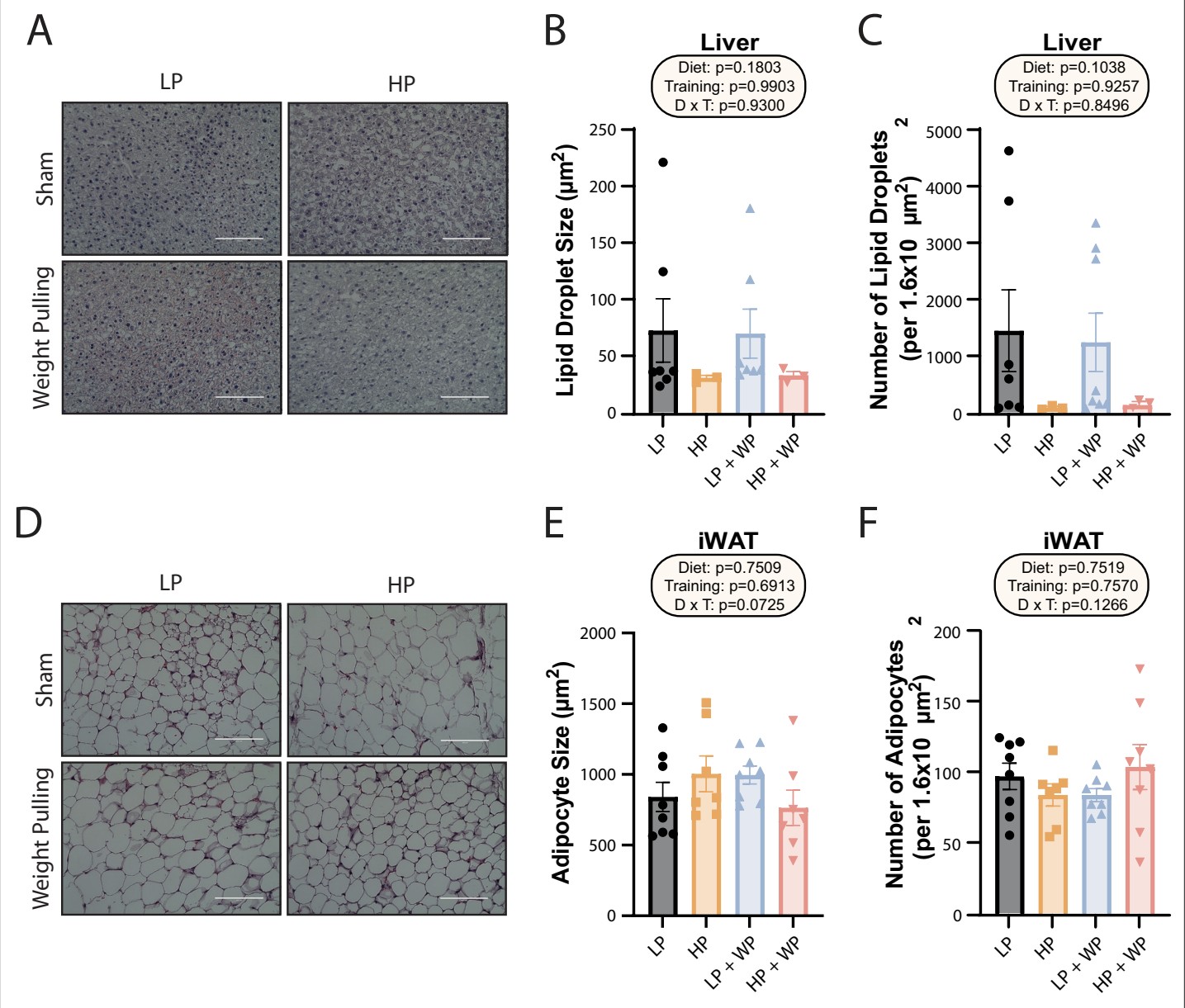

**Figure 2.** Dietary protein content and resistance training did not significantly impact liver lipid droplet or inguinal white adipocyte size. (A–C) Representative Oil-Red-O-stained liver sections from mice in the indicated groups, with quantification of average lipid droplet size (B) and number (C). n = 3–7/group. (D–F) Representative H&E-stained inguinal white adipose tissue (iWAT) sections from mice in the indicated groups, with quantification of average lipid droplet size (E) and number (F). n = 7–8/group. (B–C, E–F) Statistics for the overall effects of diet, training, and the interaction represent the p-value from a two-way ANOVA; *p<0.05, Sidak's post-test examining the effect of parameters identified as significant in the two-way ANOVA. Scale bar = 100 µM. Data represented as mean ± SEM.

The online version of this article includes the following source data for figure 2:

**Source data 1.** Dietary protein content and resistance training did not significantly impact liver lipid droplet or inguinal white adipocyte size.

during the third week of the training regimen; the maximum weight pulled by LP-fed and HP-fed mice appeared to converge after approximately 10 wk of training (*Figure 4A*). There was no overall effect of diet on the average number of sets per training bout (*Figure 4B*) or the average number of stimulatory touches by the investigator (*Figure 4C*).

The strength gains from WP did not translate into improved performance on two other tasks assessing muscle strength and function, inverted cling and rotarod (*Figure 4D–G*). We observed an overall positive effect of an LP diet on inverted cling time, an effect that persisted (p=0.0597)

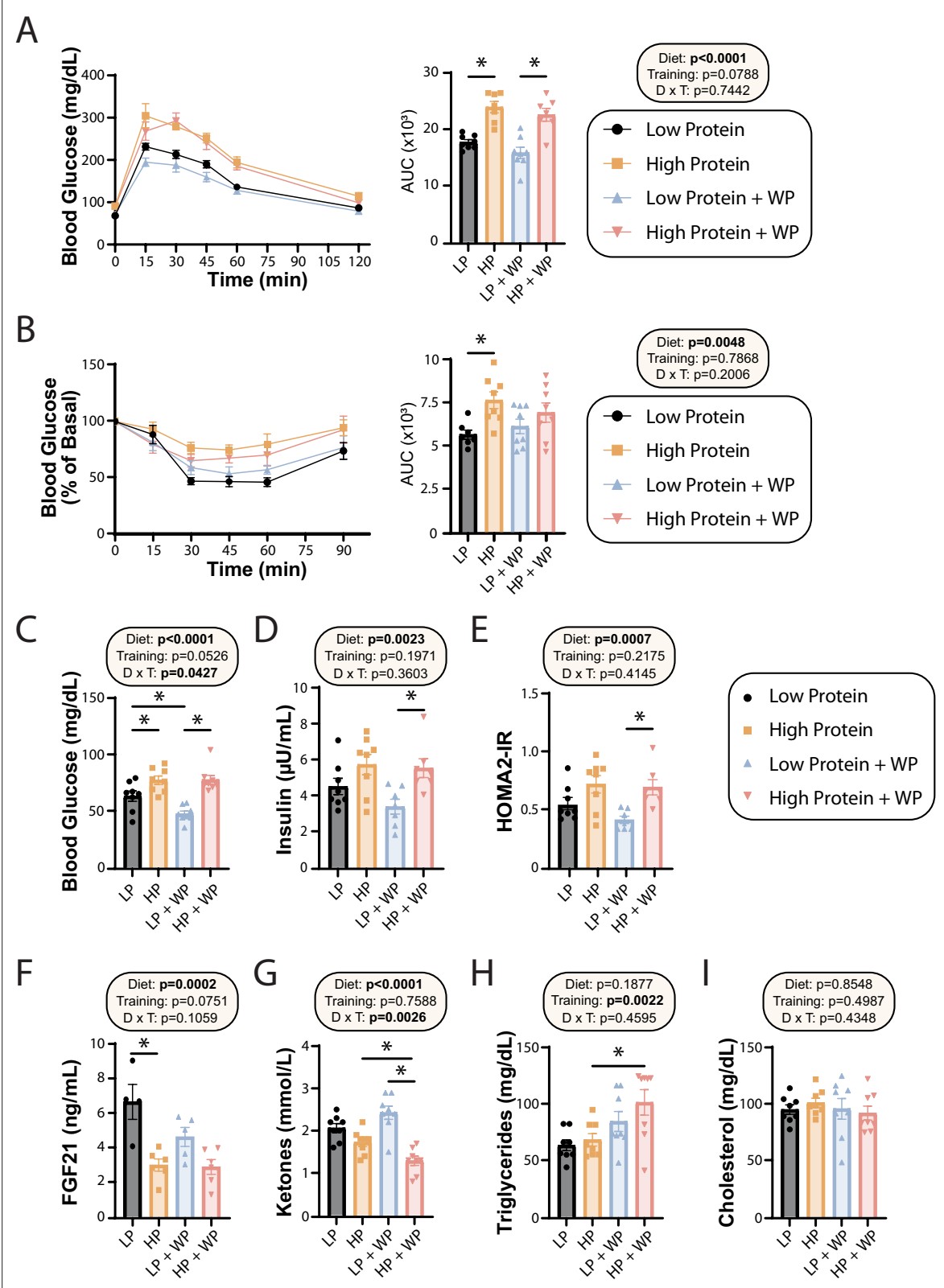

**Figure 3.** Effect of diet and exercise on glycemic control and blood metabolites. (**A, B**) Glucose (**A**) and insulin (**B**) tolerance tests were performed after 9–10 wk on the diet, respectively, and area under the curve (AUC) was calculated. n = 7–8 mice/group. (**C–E**) Blood was collected from animals after a 16 hr overnight fast; fasting blood glucose (**C**) and insulin (**D**) were determined and used to calculate HOMA2-IR (**E**). n = 7–8 mice/group. (**F–I**) Blood was collected from animals after a 16 hr overnight fast after 16 wk on the diets. Fasting FGF21 (**F**), ketones (**G**), triglycerides (**H**), and cholesterol (**I**) were

*Figure 3 continued on next page*

Figure 3 continued

determined. n = 4–8 mice/group. (**A–I**) Statistics for the overall effects of diet, training, and the interaction represent the p-value from a two-way ANOVA, *p<0.05, Sidak's post-test examining the effect of parameters identified as significant in the two-way ANOVA. Data represented as mean ± SEM.

The online version of this article includes the following source data and figure supplement(s) for figure 3:

**Source data 1.** Effect of diet and exercise on glycemic control and blood metabolites.

**Figure supplement 1.** Low-protein-fed animals have increased energy expenditure (EE) regardless of training regimen.

**Figure supplement 1—source data 1.** Low-protein-fed animals have increased energy expenditure regardless of training regimen.

in sham-exercised mice when we analyzed inverted cling performance with weight as a covariate (*Figure 4D*). Although we observed no overall effect of either diet or training on raw rotarod performance (*Figure 4F*), we observed a significant positive effect of an HP diet on the rotarod performance of sham-exercised mice when weight was considered as a covariate (*Figure 4G*).

## WP and HP diet promote muscle hypertrophy without impacting mitochondrial respiration

At the conclusion of the weight training regimen and the in vivo experiments described above, we euthanized the animals and collected numerous tissues. As noted above, WP in HP-fed mice reduced the mass of iWAT, eWAT, and liver (*Figure 1F–I*). We also collected a diverse array of skeletal muscles from different regions of the body. We have previously shown that WP induces hypertrophy in the flexor digitorum longus (FDL) (*Zhu et al., 2021*), and as we expected we found an overall positive effect of both an HP diet and WP on FDL mass, both in absolute terms and when normalized to either body mass or tibia length (*Figure 5A–C*). We observed the greatest muscle mass in mice that completed the WP regimen while consuming an HP diet, but HP diet consumption alone also had a positive effect on absolute FDL mass and FDL mass normalized to tibia length (*Figure 5A–C*).

In humans, mitochondrial respiration is increased after both aerobic and resistance training regimens (*Konopka et al., 2015*; *Konopka et al., 2019*; *McKenna et al., 2022*; *Robinson et al., 2017*), perhaps due to the high energetic demands imposed by protein synthesis (*Rolfe and Brown, 1997*; *Waterlow, 1984*), which is upregulated following resistance training (*Ogasawara et al., 2016*). However, the mitochondrial response to varying amounts of dietary protein with or without resistance training has not been examined, and we therefore performed high-resolution respirometry in permeabilized muscle fibers from the FDL. Using complex I-linked substrates pyruvate, glutamate, and malate, we found no change to leak respiration (*Figure 5D*). We next evaluated sub-saturating and saturating complex I-driven respiration by providing ADP at 0.25 mM, 0.5 mM, and 5.5 mM; none of these parameters were different in any intervention group (*Figure 5E–G*). Furthermore, we saw no difference in any group following succinate addition for complex I + II-linked respiration (*Figure 5H*).

We examined the effect of diet and WP on the mass of the quadriceps, soleus, plantaris, and the forearm flexor complex, measuring mass in absolute terms (*Figure 5—figure supplement 1A–D*) and normalized to tibia length (*Figure 5—figure supplement 1E–H*). Overall, there was no positive effect of either resistance exercise or HP diet on quadricep mass (*Figure 5—figure supplement 1A and E*), while the mass of the soleus and plantaris muscles were positively impacted by dietary protein but largely resistant to the benefits of WP (*Figure 5—figure supplement 1B–C and F–G*). However, there were strong effects of WP and a significant interaction between dietary protein and exercise on the mass of the forearm flexor complex, with the HP-fed WP mice having the greatest forearm flexor complex mass in raw weight and when normalized to tibia length (*Figure 5—figure supplement 1D and H*).

Finally, we assessed the maximum diameter of the bicep and forearm (*Figure 6A*) and found that there were clear overall effects of both dietary protein and WP on bicep and forearm diameter (*Figure 6B and C*). Bicep diameter was maximized under HP WP conditions (*Figure 6A and B*). There was a less potent but still significant effect on the forearm; HP-fed WP mice had the largest forearm diameter of all groups (*Figure 6C*). Conversely, nonexercising LP-fed mice had the smallest bicep and forearm diameter (*Figure 6A–C*).

Next, we further analyzed the FDL through staining (*Figure 7A*) and quantification (*Figure 7B–H*). There was no significant difference in mid-belly cross-sectional area (CSA) or the number of fibers per

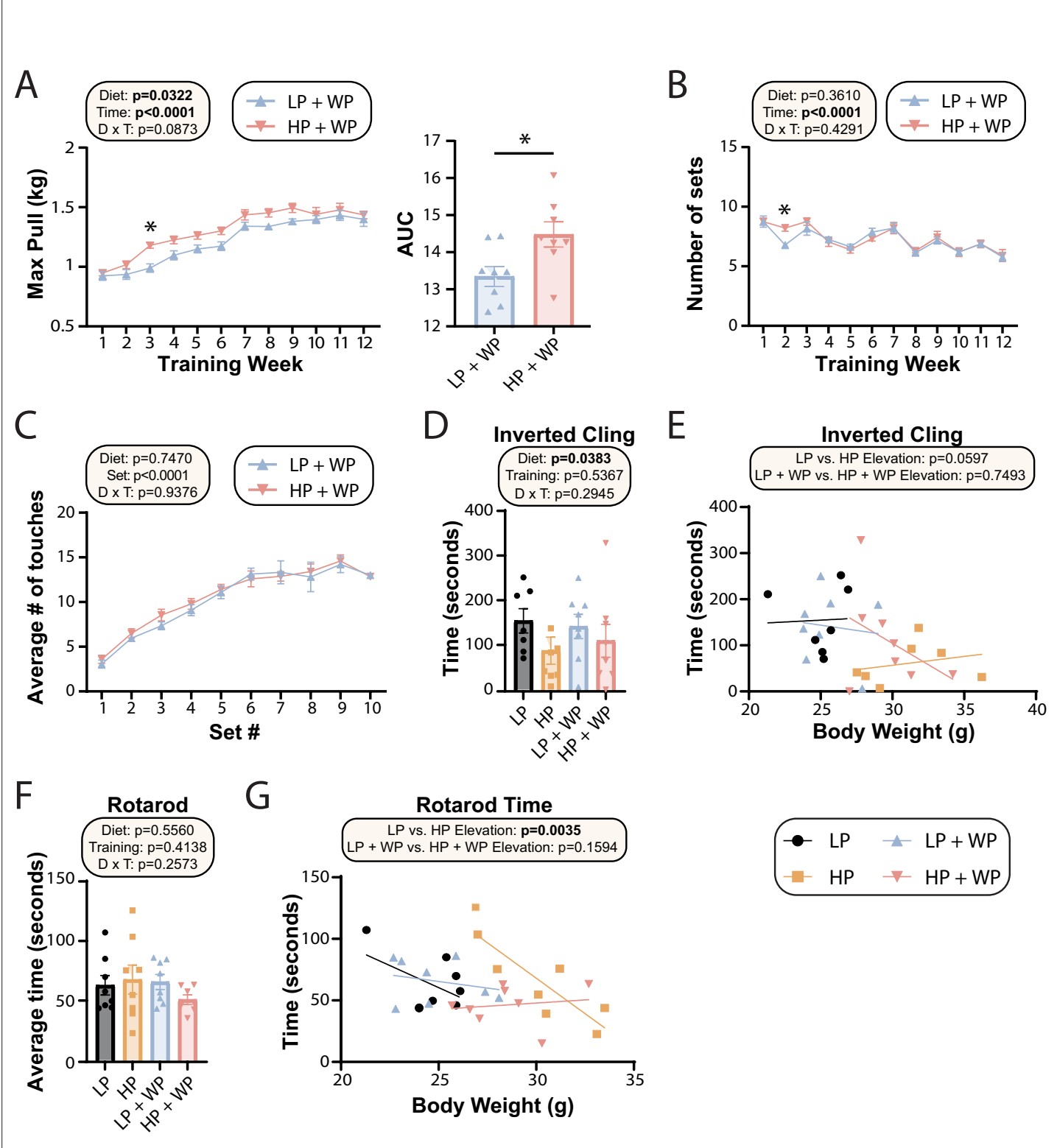

**Figure 4.** Strength and muscle growth is maximized by high protein and progressive resistance exercise. (**A–C**) Weight pulling was performed three times per week for 12 wk. The average maximum weight pulled each week with area under the curve (AUC) (**A**) and the number of sets achieved (**B**) is shown. Similarly, the number of stimulatory touches (**C**) by the investigator on all mice was averaged through multiple weeks of training and plotted per set. (**A–C**) n = 8 mice/group (**A, B**); for (**C**) values for eight mice/group were assayed over n = 5 wk. Statistics for the overall effects of diet, time or set, and the interaction represent the p-value from a two-way repeated measures (RM) ANOVA or restricted maximum likelihood (REML); *p<0.05 from

*Figure 4 continued on next page*

*Figure 4 continued*

a Sidak's post-test examining the effect of parameters identified as significant in the 2two-way ANOVA. (**D–G**) Physical performance was assessed by an inverted cling test (**D–, E**) or rotarod test (**F, G**). (**E, G**) Cling time (**E**) or rotarod time (**G**) as a function of body weight was calculated (data for each individual mouse are plotted, and slopes and intercepts were calculated using ANCOVA). n = 7–8 mice/group. Statistics for the overall effects of diet, training, and the interaction represent the p-value from a two-way ANOVA, *p<0.05, Sidak's post-test examining the effect of parameters identified as significant in the two-way ANOVA. Data represented as mean ± SEM.

The online version of this article includes the following source data for figure 4:

**Source data 1.** Strength and muscle growth is maximized by high protein and progressive resistance exercise.

cross section (*Figure 7B and C*), indicating that alterations in dietary protein or WP might not induce hyperplasia of muscle fibers (*Jorgenson and Hornberger, 2019*). There was, however, an overall effect of diet on the CSA of muscle fibers, with individual fibers in HP-fed mice having greater CSA (*Figure 7D*).

When we analyzed fiber-type-specific CSA, we observed an overall effect of diet on the size of type IIA, type IIB, and type IIX fibers, with HP diets increasing fiber size (*Figure 7F–H*). We also saw an effect of resistance exercise on the size of type IIA and type IIX fibers, with WP inducing fiber hypertrophy (*Figure 7F and H*). Finally, we observed a significant interaction between dietary protein and WP on the size of type I fibers, with the greatest hypertrophy occurring in HP-fed WP mice (*Figure 7E*).

## Discussion

Dietary protein is a critical regulator of a wide variety of biological processes that determine metabolic health and lifespan in diverse species (*Mihaylova et al., 2023*; *Trautman et al., 2022*). Consumption of protein is generally thought of as good, promoting muscle growth and strength particularly in combination with exercise, and skeletal muscle function and mass is associated with a host of health benefits including a reduced risk for diabetes and frailty (*Liao et al., 2019*; *Phillips, 2007*). Paradoxically, in sedentary humans increased consumption of dietary protein is associated with cancer, cardiovascular disease, and diabetes, as well as increased mortality (*Lagiou et al., 2007*; *Levine et al., 2014*; *Mittendorfer et al., 2020*; *Sluijs et al., 2010*; *Zhang et al., 2020*). In agreement with the potential for dietary protein to negatively impact health, LP diets and diets with reduced levels of specific essential amino acids promote healthspan and lifespan in flies and rodents (*Flores et al., 2023*; *Green et al., 2023*; *Juricic et al., 2020*; *Lee et al., 2014*; *Lees et al., 2014*; *Orentreich et al., 1993*; *Richardson et al., 2021*; *Solon-Biet et al., 2014*; *Yap et al., 2020*), and short-term protein restriction improves the metabolic health of metabolically unhealthy adult humans (*Ferraz-Bannitz et al., 2022*; *Fontana et al., 2016*).

We know that many of the humans deliberately consuming HP diets or consuming protein supplements to support their exercise regimen are not metabolically unhealthy – indeed, many of these individuals have commendable metabolic health (*Antonio et al., 2015*; *Antonio et al., 2016*; *Grøntved et al., 2012*). We considered that a potential solution to this paradox is that exercise itself is protective from the effects of HP diets; our and others previous studies examined sedentary mice and largely sedentary humans. Here, we investigated this hypothesis by feeding isocaloric diets with either high or low levels of dietary protein to mice and subjecting them to a recently validated mouse model of progressive resistance exercise, WP, or sham control exercise. Importantly, unlike aerobic exercise models like treadmill running, WP does not result in weight loss, allowing us to study the metabolic impact of exercise without confounding changes in body weight. We also took advantage of this opportunity to rigorously test long-standing assumptions regarding the effects of dietary protein and resistance exercise on muscle mass and function.

In agreement with previous studies by our lab and others, we found that in sham-exercised mice, LP-fed mice were metabolically healthier than HP-fed animals, remaining leaner and having better glycemic control; conversely, HP-fed animals accumulated fat mass and had worse glycemic control. However, when HP-fed mice were subjected to WP, the negative effects of the HP diet on overall fat mass gain as well as on individual fat depots were completely blocked. Surprisingly, although WP did further lower fasting blood glucose in LP-fed mice and these mice had a tendency toward improved glucose tolerance and lower fasting insulin, the negative effect of the HP diet on glucose tolerance and insulin sensitivity was not blocked by WP.

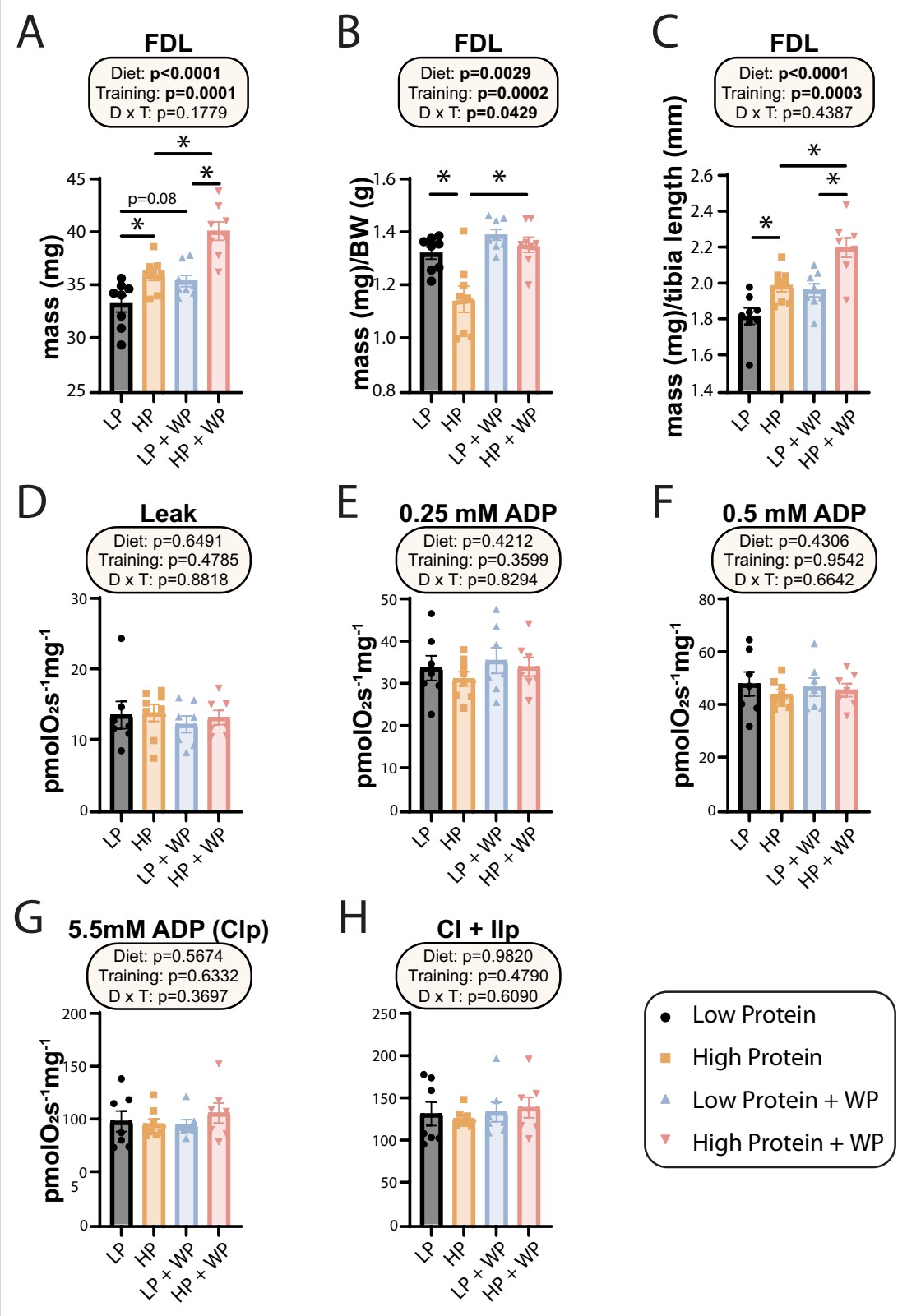

**Figure 5.** Weight pulling and high-protein diet increased flexor digitorum longus (FDL) mass but not mitochondrial respiration. (A–C) The muscle mass of the FDL in absolute mass (A), normalized to body weight (B), and (C) normalized to tibia length. n = 8/group. (D–H) Mitochondrial respiration parameters as measured in the FDL, including mitochondrial leak (D), and following addition of 0.25 mM (E), 0.5 mM (F), and 5.5 mM ADP (G). (H) 10 mM succinate was provided to evaluate complex I and II-driven respiration. n = 7–8 mice/group. (A–H) Statistics for the overall effects of diet, training, and

*Figure 5 continued on next page*

*Figure 5 continued*

the interaction represent the p-value from a two-way ANOVA, *p<0.05, Sidak's post-test examining the effect of parameters identified as significant in the two-way ANOVA. Data represented as mean ± SEM.

The online version of this article includes the following source data and figure supplement(s) for figure 5:

**Source data 1.** Weight pulling and high-protein diet increased flexor digitorum longus (FDL) mass but not mitochondrial respiration.

**Figure supplement 1.** Weight pulling and high-protein diet increase the mass of specific muscles.

**Figure supplement 1—source data 1.** Weight pulling and high-protein diet increase the mass of specific muscles.

While HP-feeding did, as we expected, support muscle mass gain and strength gain more than an LP diet, the superiority of the HP diet in the maximum weight pulled group was transitory; by the end of the training period, LP-fed mice could pull just as much weight as HP-fed mice. There was a modest effect of diet on the performance of mice on inverted cling and rotarod tasks, with HP-fed sham mice performing better on the rotarod than LP-fed sham mice when weight was considered as a covariate, but this effect was not observed in WP groups. As rotarod and inverted cling outcomes can be improved by voluntary aerobic exercise (wheel running) (*Graber et al., 2015*), this could be the result of either the type of exercise or the precise muscles strengthened by WP vs. aerobic exercise. Somewhat surprisingly, mitochondrial respiration outcomes were also independent of diet and exercise in the FDL, but we did not examine mitochondrial function in other tissues.

Our study was subject to a number of limitations. We examined only young inbred C57BL/6J male mice, and we have previously shown that sex, genetic background, and age impact the metabolic response to dietary protein (*Green et al., 2022*). We consider it likely that these factors may also influence the response to exercise, and examining the interaction between diet and WP in females, older mice, and different genetic backgrounds should be a high priority for future studies. Further, as the metabolic response to dietary protein is responsive to the exact level of protein in the diet, examining additional levels of dietary protein between the LP and HP diets we examined here might provide greater insight. This would also allow the study of dietary protein levels within the acceptable

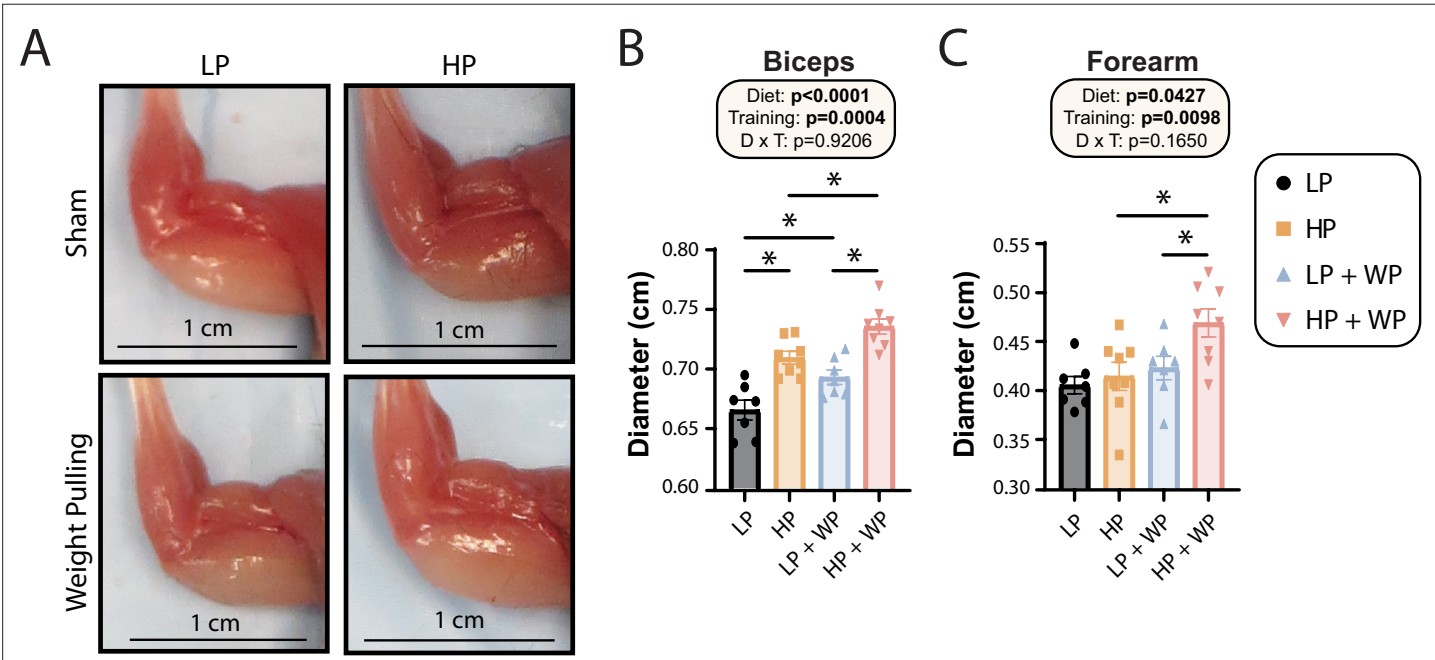

**Figure 6.** Bicep and forearm hypertrophy is maximized by high-protein (HP) diets and weight pulling (WP). (A–C) Representative images of arm musculature (A) with quantification of biceps (B) and forearm (C) diameter. (B, C) n = 7–8 mice per group. Statistics for the overall effects of diet, training, and the interaction represent the p-value from a two-way ANOVA, *p<0.05, Sidak's post-test examining the effect of parameters identified as significant in the two-way ANOVA. Data represented as mean ± SEM.

The online version of this article includes the following source data for figure 6:

**Source data 1.** Bicep and forearm hypertrophy is maximized by high-protein (HP) diets and weight pulling (WP).

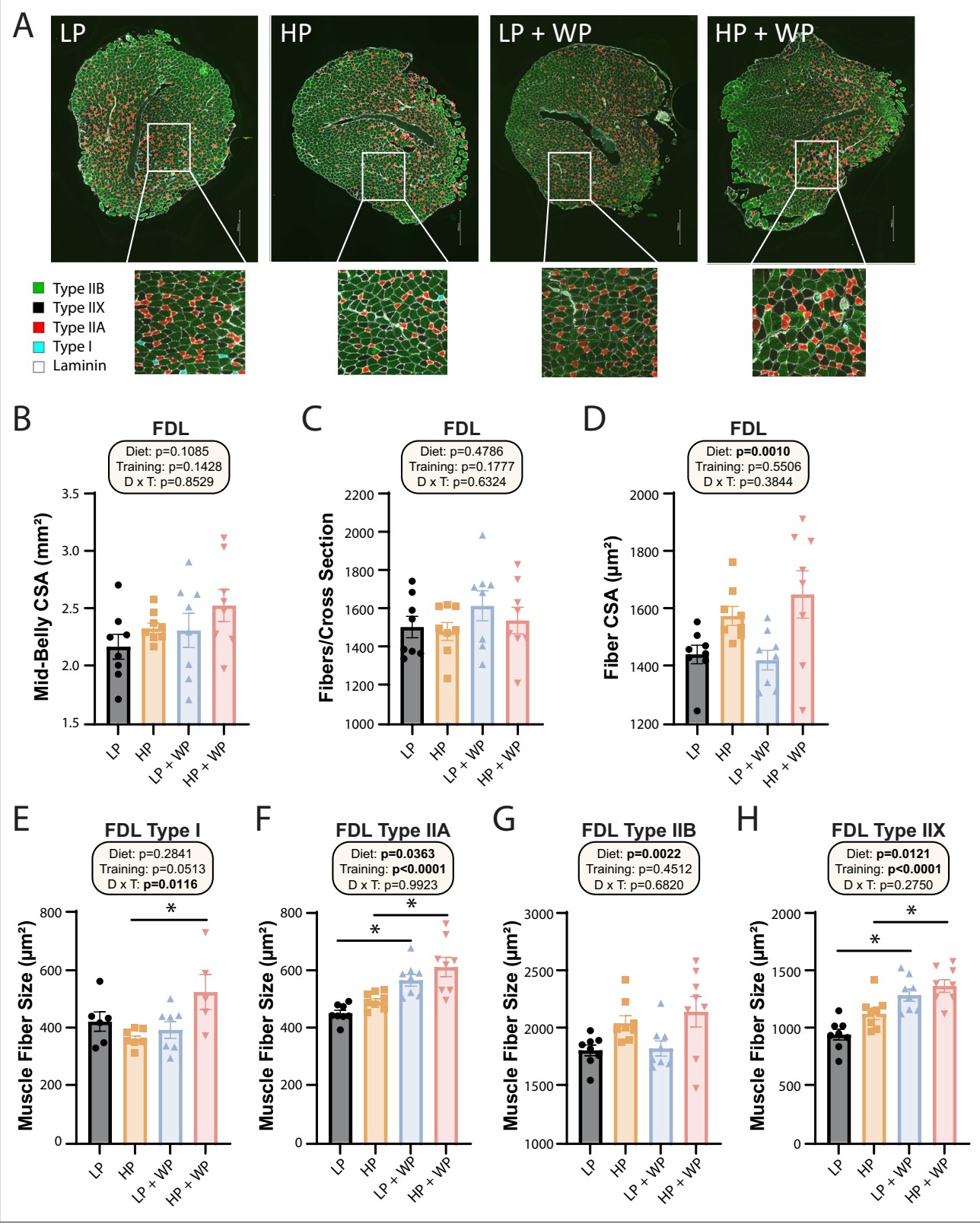

**Figure 7.** Flexor digitorum longus (FDL) fiber-type hypertrophy is maximized by high-protein diets and weight pulling. (**A–H**) Representative images of the FDL and fiber type with quantification of mid-belly cross-sectional area (CSA) (**B**), fibers per cross section (**C**), fiber CSA (**D**), and individual muscle fiber type size: type I (**E**), type IIA (**F**), type IIB (**G**), and type IIX (**H**).n = 5–8 mice per group. Statistics for the overall effects of diet, training, and the

*Figure 7 continued on next page*

*Figure 7 continued*

interaction represent the p-value from a two-way ANOVA, *p<0.05, Sidak's post-test examining the effect of parameters identified as significant in the two-way ANOVA. Data represented as mean ± SEM.

The online version of this article includes the following source data for figure 7:

**Source data 1.** Flexor digitorum longus (FDL) fiber-type hypertrophy is maximized by high-protein diets and weight pulling.

macronutrient distribution range for humans, which is 10–35% of calories from protein. Finally, to keep our diets isocaloric, isonitrogenous, and consistent in fat content, we had to reduce carbohydrates in the HP diet, which might play a role in the results found here.

Many people who eat HP diets or take protein supplements also engage in aerobic exercise, which we did not examine in the present study. Here, we have not examined the effects of dietary protein and exercise on molecular changes and have not explained how WP inhibits HP-induced gains in adiposity. The hormone FGF21, which is induced by an LP diet, is responsive to exercise in mice and humans (*Haghighi et al., 2022*; *Jin et al., 2022*; *Li et al., 2022*), but we did not observe increased FGF21 in the LP weight-trained group, suggesting another mechanism could be at play. As for the improved body composition of HP-fed WP mice, we hypothesize that these interventions may activate beiging in iWAT or BAT. Other possibilities include changes in mitochondrial respiration in muscles other than the FDL or exercise-induced alterations in lipogenesis or lipolysis. Lastly, we have not explored the role of individual amino acids in the response; many athletes and body builders take supplements enriched in BCAAs, and future work should determine whether there is a specific role for these or other dietary amino acids in the metabolic and physiological response to resistance exercise.

Finally, when broken down to individual muscle fiber types, we saw an increase in type 1 muscle fiber size in the FDL of the HP + WP group, which indicates that both HP and resistance training are necessary for inducing hypertrophy of slow-twitch, oxidative fiber types. However, as three of our HP + WP samples had no detectable type 1 fibers, future research should evaluate this effect in muscles with an abundance of type 1 fibers. These findings are consistent with specific fiber-type hypertrophy previously observed in the FDL in response to WP (*Zhu et al., 2021*). We saw a similar induction of growth by both diet and training in type IIA fibers, which are fast-twitch, oxidative, and glycolytic. Here, HP feeding resulted in larger fibers than LP in the context of both sham training and WP, which is consistent with human recommendations that increased dietary protein supports muscle hypertrophy (*Thomas et al., 2016*). Surprisingly, there was no effect of training observed in fast-twitch, glycolytic-type IIB fibers, but there was an effect of diet, with HP feedings inducing hypertrophy. Finally, fast-twitch-type IIX fibers demonstrated a similar trend as type IIA fibers, with both an HP diet and WP inducing growth. A striking takeaway observed here is that there was no difference in hypertrophy of type IIX and IIA fibers in the FDL induced by WP when fed an LP or HP diet. Though these changes occurred in just one muscle, and effects on other muscles may vary, our results indicate that both resistance training and HP diets individually and in combination optimize growth of muscle fibers.

In summary, dietary protein and progressive resistance exercise interact to regulate metabolic health as well as strength and muscle mass. Specifically increasing protein consumption has – as we anticipated based on both animal and human data – negative effects on the metabolic health of mice that are not exercising. However, the negative effects of an HP diet on body composition are eliminated in mice performing progressive resistance exercise training. While mice consuming an HP diet gained strength during WP more quickly than LP-fed mice, the ultimate max load the HP and LP-fed mice could pull by the end of 12 wk was indistinguishable despite clear differences in muscle mass. Future research could investigate this apparent paradox by examining the effect of the diet and WP regimens on muscle mass and fiber type at earlier time points than those examined in this study. However, our results are consistent with studies in healthy, resistance-trained individuals that show HP diets can be consumed without negative effects on body composition (*Antonio et al., 2015*; *Antonio et al., 2016*). We hypothesize that further studies will find that sedentary individuals would be metabolically healthier if consuming an LP diet. Finally, our results suggest that a precision nutrition approach to metabolic health must consider not only diet, sex, and genetic variation, but also take into account activity level.

# Methods

**Key resources table**

| Reagent type (species) or resource | Designation | Source or reference | Identifiers | Additional information |
|---|---|---|---|---|
| Strain, strain background (*Mus musculus*), male | C57BL/6J | The Jackson Laboratory | Cat# JAX:000664; RRID:IMSR_JAX:000664 | |
| Antibody | Anti-laminin | Millipore Sigma | #L9393 | 1:500 |
| Antibody | Myosin heavy chain type I | Developmental Studies Hybridoma Bank | #BA-D5-s | 1:100 |
| Antibody | Myosin heavy chain type IIA | Developmental Studies Hybridoma Bank | #SC-71-s | 1:100 |
| Antibody | Myosin heavy chain type IIB | Developmental Studies Hybridoma Bank | #BF-F3-s | 1:10 |
| Antibody | Rabbit IgG (H+L) Cross-Adsorbed Secondary Antibody in ICC/IF | Invitrogen | #A11011 | 1:5000 |
| Antibody | Alexa Fluor 647 AffiniPure Goat Anti-Mouse IgG, Fcγ subclass 2b specific | Jackson ImmunoResearch | #115-605-207 | 1:100 |
| Antibody | Alexa Fluor 488 AffiniPure Goat Anti-Mouse IgG, Fcγ subclass 1 specific | Jackson ImmunoResearch | #115-545-205 | 1:3000 |
| Antibody | Goat anti-Mouse IgM (Heavy chain) Cross-Adsorbed Secondary Antibody, Alexa Fluor 350 | Invitrogen | #A-31552 | 1:500 |
| Antibody | ProLong Gold Antifade Mountant | Invitrogen | #P36930 | |
| Commercial assay or kit | Mouse/Rat FGF21 ELISA | R&D Systems | #MF2100 | |
| Commercial assay or kit | Mouse insulin ELISA | Crystal Chem | #90080 | |
| Commercial assay or kit | Cholesterol | Pointe Scientific | #23-666-200, #23-666-202, #23-666-201 | |
| Commercial assay or kit | Triglycerides | Pointe Scientific | #23-666-411, #23-666-410, #23-666-412 | |

## Animal care, housing, and diet

All procedures were performed in conformance with institutional guidelines and were approved by the Institutional Animal Care and Use Committee of the William S. Middleton Memorial Veterans Hospital. Wild-type male C57BL/6J mice were procured from the Jackson Laboratory (000664) at 6 wk of age. All studies were performed on animals or on tissues collected from animals. All mice were acclimated to the animal research facility for at least 1 wk before entering studies. All animals were housed two per cage in static microisolator cages in a specific pathogen-free mouse facility with a 12:12 hr light–dark cycle, maintained at approximately 22°C. At the start of the experiment, mice were randomized into weight-equivalent groups at the cage level to receive either the 7% (LP, TD.140712) or 36% (HP, TD.220097) amino acid-defined diets and assigned to the sham or WP groups; all diets were obtained from Envigo. In the HP diet, carbohydrates were reduced and amino acids increased relative to the LP diet in order to keep the diets isocaloric, while calories from fat were held fixed. Full diet descriptions, compositions, and item numbers are provided in *Table 1*.

## Weight training paradigm

The procedures of this WP regimen (acclimation, weighted paradigm, and unweighted paradigm) were described previously (*Zhu et al., 2021*).

## Metabolic phenotyping

Glucose and insulin tolerance tests were performed following a 16 hr overnight or 4 hr fast, respectively, and then injecting either glucose (1 g/kg) or insulin (0.75 U/kg) intraperitoneally (*Bellantuono et al., 2020*; *Yu et al., 2019*). Glucose measurements were taken using a Bayer Contour blood glucose meter and test strips. Blood ketone measurements were taken using a KetoBM ketone meter and test

strips that detect beta-hydroxybutyrate, the most abundant ketone in the body and proxy for ketone level. Mouse body composition was determined using an EchoMRI Body Composition Analyzer. For assays of multiple metabolic parameters ($O_2$, $CO_2$, food consumption, and activity tracking), mice were acclimatized to housing in a Columbus Instruments Oxymax/CLAMS-HC metabolic chamber system for approximately 24 hr, and data from a continuous 24 hr period was then recorded and analyzed.

## Assays and kits

Unless otherwise noted, all kits were conducted using samples from overnight fasted mice. Circulating FGF21 was determined using Quantikine's Mouse/Rat FGF21 ELISA Kit (#MF2100).

Circulating insulin was determined using Crystal Chem's Ultra Sensitive Mouse Insulin ELISA Kit (#90080). Circulating Triglycerides and cholesterol were determined using Pointe Scientific's Triglycerides Liquid Reagents (#23-666-410, #23-666-411, #23-666-412) and Pointe Scientific Cholesterol Liquid Reagents (#23-666-200, #23-666-201, #23-666-202). Protocols for these kits can be found at the manufacturer. All samples were run in duplicates.

## Tissue collection

Terminal collections were performed 60–96 hr after the final training session and after a 3 hr morning fast. During this procedure, all potentially identifiable information (e.g., tail markings, etc.) was masked, and then the animals were weighed and subsequently anesthetized with isoflurane. The plantaris, FDL, forearm flexor complex (FF) (which consisted of the flexor carpi ulnaris, flexor carpi radialis, flexor digitorum superficialis, and all three heads of the flexor digitorum profundus), and soleus (SOL) muscles from both the left and right hindlimb were then weighed. The left and right plantaris and right FDL were submerged in optimal cutting temperature compound (OCT, Tissue-Tek; Sakura Finetek, The Netherlands) at resting length and frozen in liquid nitrogen-chilled isopentane. Left FDL was prepared for mitochondrial analysis. At this point, the mice were euthanized by cervical dislocation, a photograph that included both a scale bar and the musculature of the left forelimb was obtained, and then quadriceps muscles from both the left and right sides of the body were collected. In addition to skeletal muscles, the iWAT, BAT, eWAT, liver, as well as the right tibia bone, were collected post mortem and frozen in liquid nitrogen or prepared for histology (OCT or formalin, to then be incubated in ethanol). Oil-Red-O staining of liver samples and H&E staining of iWAT samples were conducted by the UW Carbone Cancer Center Experimental Animal Pathology Lab (UWCCC EAPL) on a fee-for-service basis. Muscle photographs were taken with a ruler for scale, and precise maximum muscle diameter was measured using ImageJ. All of the collection procedures were performed by investigators blinded to treatment and diet group.

## Immunohistochemistry and image analysis

Fiber-type staining and analysis was performed as previously described (*Zhu et al., 2021*). ImageJ software was utilized to quantify hepatic lipid droplets, and the ImageJ Adiposoft plugin was used to quantify iWAT adipocytes.

## High-resolution respirometry

FDL muscle was excised, weighed, and cut in half. The distal half was snap-frozen in liquid nitrogen while proximal half was placed in ice-cold (4°C) Buffer X containing (in mM) 7.23 K2EGTA, 2.77 CaK2EGTA, 20 imidazole, 20 taurine, 5.7 ATP, 14.3 phosphocreatine, 6.56 $MgCl_2 \cdot 6H_2O$, and 50 K-MES (pH 7.1). Muscle was mechanically permeabilized with fine tip forceps removing connective and adipose tissue. Fiber bundles were chemically permeabilized in Buffer X with saponin (50 µg/mL) for 30 min on ice. Permeabilized muscle fiber bundles were rinsed with MiR05 containing (in mM) 0.5 mM EGTA, 3 $MgCl_2$, 60 K-lactobionate, 20 taurine, 10 $KH_2PO_4$, 20 HEPES, 110 sucrose, 1 g/L BSA essentially fatty acid free (pH 7.1), and 25 µM blebbistatin. Fibers were then blotted on filter paper to remove excess buffer and weighed on a microscale. Fibers bundles (1.5–2.5 mg) were then placed into chambers of the Oxygraph-2k (O2K; Oroboros Instruments, Innsbruck, Austria) containing MiR05 plus 12.5 µM blebbistatin, a myosin ATPase inhibitor, at 37°C. Chambers were hyperoxygenated to ~425 uM, and mitochondrial respiration was supported by complex I-linked substrates 5 mM pyruvate, 10 mM glutamate, and 0.5 mM malate. ADP was provided at 0.25 mM, 0.5 mM, and 5.5 mM to evaluate sub-saturating and saturating complex I-driven mitochondrial respiration. Mitochondrial

membrane integrity was evaluated using 10 µM cytochrome *c*. A >15% change in oxygen consumption was considered a loss of mitochondrial membrane integrity and three samples were removed from final analysis. A 10 mM succinate was provided to evaluate complex I and II-driven respiration. Data is normalized to muscle fiber bundle wet weight.

## Statistics

Data are presented as the mean ± SEM unless otherwise specified. Statistical analyses were performed using one-way, two-way ANOVA, two-way repeated measures (RM) ANOVA, or a mixed-effects model (restricted maximum likelihood [REML]), followed by an appropriate posttest as specified in the figure legends. In all figures, n represents the number of biologically independent animals, except in *Figure 5C*, where n represents the number of weeks that touches were determined for across all animals. Outliers were determined using GraphPad Prism Grubbs' calculator (https://www.graphpad.com/quickcalcs/grubbs1/). Sample sizes were chosen based on our previously published experimental results with the effects of dietary interventions plus consideration of the training time required by personnel (*Cummings et al., 2018*; *Fontana et al., 2016*; *Yu et al., 2021*; *Yu et al., 2018*). Data distribution was assumed to be normal, but this was not formally tested. Inclusion and exclusion criteria were established prior to the experiment. All data were included besides outliers or failed analysis.

## Reporting

This article has been submitted with the ARRIVE guidelines 2.0 Essential 10 Checklist.

## Acknowledgements

We thank Santosh Kumari for assistance with tissue collection. The Lamming Laboratory is supported in part by the NIH/National Institute on Aging (AG056771, AG062328, AG061635, AG081482, and AG084156), NIH/NIDDK (DK125859), and startup funds from the University of Wisconsin-Madison School of Medicine and Public Health and Department of Medicine to DWL. The Konopka lab is supported in part by the NIH/NIA (AG067464, AG076941, and AG081482). The Hornberger lab is supported in part by the NIH/NIAMS (AR057347 and AR074932). MET is supported by an NIH/NIA F99/K00 fellowship (AG083290). MMS is supported by a Research Supplement to Promote Diversity in Health-Related Research (3RF1AG056771-06S1). CYY is supported by a NIH/NIA F32 postdoctoral fellowship (AG077916). The UWCCC Experimental Pathology Laboratory is supported by the University of Wisconsin Carbone Cancer Center Support Grant P30 CA014520. The Lamming laboratory was supported in part by the U.S. Department of Veterans Affairs (I01-BX004031 and IS1-BX005524), and this work was supported using facilities and resources from the William S Middleton Memorial Veterans Hospital. The content is solely the responsibility of the authors and does not necessarily represent the official views of the NIH. This work does not represent the views of the Department of Veterans Affairs or the United States Government.

## Additional information

### Competing interests

Dudley W Lamming: has received funding from, and is a scientific advisory board member of, Aeovian Pharmaceuticals, which seeks to develop novel, selective mTOR inhibitors for the treatment of various diseases. The other authors declare that no competing interests exist.

### Funding

| Funder | Grant reference number | Author |
|---|---|---|
| National Institute on Aging | AG056771 | Dudley W Lamming |
| National Institute on Aging | AG067464 | Adam R Konopka |

| Funder | Grant reference number | Author |
| --- | --- | --- |
| National Institute of Arthritis and Musculoskeletal and Skin Diseases | AR057347 | Troy A Hornberger |
| National Institute on Aging | AG083290 | Michaela E Trautman |
| National Institute on Aging | AG077916 | Chung-Yang Yeh |
| National Institute on Aging | 3RF1AG056771-06S1 | Michelle M Sonsalla |
| National Institute on Aging | AG062328 | Dudley W Lamming |
| National Institute on Aging | AG061635 | Dudley W Lamming |
| National Institute on Aging | AG081482 | Dudley W Lamming |
| National Institute on Aging | AG084156 | Dudley W Lamming |
| National Institute on Aging | AG076941 | Adam R Konopka |
| National Institute on Aging | AG081482 | Adam R Konopka |
| National Institute of Arthritis and Musculoskeletal and Skin Diseases | AR074932 | Troy A Hornberger |

The funders had no role in study design, data collection and interpretation, or the decision to submit the work for publication.

## Author contributions

Michaela E Trautman, Leah N Braucher, Conceptualization, Formal analysis, Investigation, Writing – original draft, Writing – review and editing; Christian Elliehausen, Formal analysis, Investigation, Writing – review and editing; Wenyuan G Zhu, Formal analysis, Investigation, Methodology, Writing – review and editing; Esther Zelenovskiy, Madelyn Green, Michelle M Sonsalla, Chung-Yang Yeh, Investigation; Troy A Hornberger, Adam R Konopka, Conceptualization, Funding acquisition, Project administration, Writing – review and editing; Dudley W Lamming, Conceptualization, Formal analysis, Funding acquisition, Writing – original draft, Project administration, Writing – review and editing

## Author ORCIDs

Michaela E Trautman http://orcid.org/0000-0002-3172-0436
Troy A Hornberger http://orcid.org/0000-0002-2349-1899
Dudley W Lamming https://orcid.org/0000-0002-0079-4467

## Ethics

All animal procedures conducted at the William S. Middleton Memorial VA Hospital were approved by the Institutional Animal Care and Use Committee of the William S. Middleton Memorial Veterans Hospital (Assurance ID: D16-00403). All animals were housed and cared for in an Association for Assessment and Accreditation of Laboratory Animal Care (AAALAC)-accredited animal facility under specific pathogen-free conditions.

Reviewer #1 (Public Review): https://doi.org/10.7554/eLife.91007.3.sa1
Reviewer #2 (Public Review): https://doi.org/10.7554/eLife.91007.3.sa2
Author Response https://doi.org/10.7554/eLife.91007.3.sa3

# Additional files

## Supplementary files

• MDAR checklist

## Data availability

All data generated or analyzed during this study are included in the manuscript and supporting files.

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
