## [Editor Report · eLife assessment]

This study presents a **valuable** finding on the relationship between high-protein diet and resistance exercise on fat accumulation and glucose homeostasis. The evidence supporting the claims of the authors is **solid**, although the inclusion of mechanistic insight would have strengthened the study. The work will be of interest to dietician and exercise biologists working to understand the synergy between diet and physical activity.

---

## [Referee Report · Reviewer #1 (Public Review)]

Summary:

The study conducted on mice establishes a noteworthy connection between dietary protein intake and resistance exercise impact on metabolic health and muscle development. In sedentary mice, a diet rich in protein resulted in excessive fat accumulation and compromised blood sugar regulation in comparison to a diet low in protein. Intriguingly, when mice followed the high protein diet alongside progressive resistance training, they exhibited protection against surplus fat gain, though blood glucose regulation remained impaired. The research also revealed that resistance training notably enhanced muscle hypertrophy induced by exercise, particularly in mice on the high protein diet. Although the maximum strength achieved was similar across diets, this highlights the potential synergy between high protein consumption and resistance exercise in promoting skeletal muscle growth.

Strengths:

The study possesses several significant strengths. Firstly, it combines controlled dietary manipulations with resistance exercise, providing a comprehensive understanding of their combined effects on metabolic health and muscle growth. The use of mouse models, while not directly translatable to humans, offers a controlled experimental environment, enabling precise measurements and observations. Moreover, the study reveals nuanced outcomes such as the differential impact of high protein intake on adiposity and muscle hypertrophy. The emphasis on both positive and negative findings lends balance to the conclusions, enhancing the overall credibility of the study. Additionally, the clear delineation of diet-exercise interactions contributes to the broader understanding of dietary and exercise recommendations for metabolic health and muscle development.

Weaknesses:

Certain limitations warrant consideration. Firstly, the study's exclusive reliance on mice might limit the generalizability of the findings to humans due to inherent physiological differences. Additionally, the absence of direct investigation into the underlying molecular mechanisms responsible for the observed outcomes leaves room for speculation. Moreover, the research's concentration on male and young mice raises questions about the applicability of these findings to female and older subjects. Lastly, the study's duration and the specific resistance exercise protocol utilized might not fully reflect long-term human scenarios, underscoring the need for further research in more diverse populations and over extended timeframes.

---

## [Referee Report · Reviewer #2 (Public Review)]

Summary:

In this manuscript, Trautman et al. set out to test the hypothesis that increased intake of dietary protein is deleterious to health when uncoupled from resistance training.

Strengths:

The experimental design is well crafted and the experiments provide useful information supporting the hypothesis. The authors take into account the limitations of their study in the discussion, and guide the reader through their results and the interpretation in a fair and measured way, without overstating claims.

Weaknesses:

As acknowledged by the authors in the discussion section, this study only features a small sample of male mice from a single strain. Thus the results may not hold when female mice and diverse genetic backgrounds are analyzed. The lack of repeated measures of physiological parameters is also a limitation of the study. Measurements of body weight, body composition, food (calorie) consumption, and locomotor/strength assays could have been provided throughout the study and compared to a baseline value for each animal.

---

## [Author Response]

The following is the authors’ response to the original reviews.

We would first like to thank the reviewers for their time and effort in their critical review of our manuscript, and appreciate the opportunity to address these comments. We thank the reviewers for appreciating that our experimental design is well crafted, and contributes to the broader understanding of dietary exercise recommendations for metabolic health and muscle development. We have revised the figures and text in accordance with the reviewer’s recommendations, and hope that they appreciate the revised version.

**Reviewer #1:**
1. A significant limitation of this study pertains to the absence of a detailed exploration into the mechanistic underpinnings of the interaction between high protein intake and resistance exercise at the molecular level. The authors should provide a comprehensive discussion on potential avenues or prospective research directions to address this gap in understanding.

We agree and have added some theories in the discussion on page 14.

1. Figure 4 and Figure 7 can be moved to supplementary and text in the description can be arranged accordingly to make a better flow of the story.

We agree with this suggestion and have made adjustments.

1. The authors have used a high protein diet (36% calorie from protein) and a low protein diet (7% calorie from protein) for this study. The authors should explain whether this mouse diet is practically comparable to the human's high protein (2% of BW) and low protein diet (less than 0.8% BW) or not.The high protein diet is comparable to a human diet of 180 grams of protein ((0.36x2000 calories)/4 calories per gram=180 g), which is in a range that some people consume, particularly bodybuilders and athletes. The low protein diet is equivalent to 35 grams of protein per day ((0.07x2000 calories)/4 calories/gram=35g), and a diet of just 7% protein is not recommended for humans per the Acceptable Macronutrient Distribution Range (AMDR) of 10-35% dietary protein set by the Institute of Medicine (IOM). We have addressed this on page 14.1. The color coding of the error bar and lines does not match with the group description in almost every figure. Maybe the authors could choose more contrasting colors.

Thanks, we have adjusted the coloring of the error bars and lines in all figures.

1. In Figure 3C-E it seems like the number of biological samples is not consistent in the LP+WP group. If the authors have excluded any outlier from the analysis, that should be included in the methodology.

We did list outliers in the methodology in the statistics section (page 19): “Outliers were determined using GraphPad Prism Grubbs’ calculator (https://www.graphpad.com/quickcalcs/grubbs1/).”

**Reviewer #2:**
Very nice work! I do not have a whole lot to say in terms of experiments, analysis, or data to present other than what is in my public review (and you cannot really provide it as it was not in the experimental design). The manuscript is also very well written. My only question is about the following two sentences in the introduction:

"Both exercise and amino acids activate the mechanistic target of TOR (mTOR) protein kinase, which stimulates the protein synthesis machinery needed to stimulate skeletal muscle hypertrophy (Schiaffino et al., 2021). Therefore, The Academy of Nutrition and Dietetics recommends consuming 1.2-2.0 grams of protein per kg of body weight (BW) per day in physically active individuals (Thomas et al., 2016)." I am not sure how the second sentence follows from the first, so I am not convinced that "therefore" is the right adverb in the right place.

Thanks for pointing this out. We have added a clarifying transition to the text (page 3).